# Do Depth-Grown Models Overcome the Curse of Depth? An In-Depth Analysis

**Ferdinand Kapl**[1,2,‡] **Emmanouil Angelis**[1,2,‡] **Tobias Höppe**[1,2,‡]
**Kaitlin Maile**[3,†] **Johannes von Oswald**[3,†] **Nino Scherrer**[3,†] **Stefan Bauer**[1,2,†]
[1]Technical University of Munich   [2]Helmholtz AI, Munich   [3]Google, Paradigms of Intelligence

## Abstract

Gradually growing the depth of Transformers during training can not only reduce training cost but also lead to improved reasoning performance, as shown by MIDAS (Saunshi et al., 2024). Thus far, however, a mechanistic understanding of these gains has been missing. In this work, we establish a connection to recent work showing that layers in the second half of non-grown, pre-layernorm Transformers contribute much less to the final output distribution than those in the first half—also known as the *Curse of Depth* (Sun et al., 2025; Csordás et al., 2025). Following a hypothesis–experiment–evidence based analysis, we demonstrate that growth via gradual middle stacking yields more effective utilization of model depth, alters the residual stream structure, and facilitates the formation of permutable computational blocks. In addition, we propose a lightweight modification of MIDAS that yields further improvements in downstream reasoning benchmarks. Overall, this work highlights how the gradual growth of model depth can lead to the formation of distinct computational circuits and overcome the limited depth utilization.

## 1 Introduction

Large language models (LLMs) have achieved impressive capabilities, but at high computational and energy cost. Much of this progress has relied on increasing model parameters, particularly across the depth dimension (Kaplan et al., 2020; Hoffmann et al., 2022). Yet, growing evidence shows that in modern pre-layer Transformers, deeper layers contribute little to final performance, a phenomenon termed the *Curse of Depth* (Yin et al., 2024; Gromov et al., 2025; Li et al., 2025; Men et al., 2025; Sun et al., 2025; Csordás et al., 2025). Separately, *gradual growth* methods were introduced primarily as a way to reduce training compute by starting with a smaller model and expanding it during training, reusing learned weights across stages (Gong et al., 2019; Reddi et al., 2023). A notable example is MIDAS (Saunshi et al., 2024), which increases depth by inserting new layers into the middle of the network. While MIDAS can speed up training, it has also been reported to improve reasoning performance.

In this work, using the hypothesis–experiment–evidence analysis paradigm, we show that gradual depth growth is not only compute-efficient during training, but it can also counteract the Curse of Depth by changing how computation is distributed across layers (Figure 1). Concretely, we: (i) reproduce MIDAS on SmolLM-v1 backbones (360M, 1.7B), confirming improved reasoning and a $1.29\times$ training speedup over a non-grown baseline; (ii) propose LIDAS, a simple layer-duplication growth strategy that matches or exceeds MIDAS and conventional training on reasoning benchmarks without degrading NLL or knowledge performance; and (iii) provide mechanistic evidence that grown models use depth more effectively and develop structured, permutable computation in the middle of the network.

---

[*]Equal contribution.
[†]Provided equal in-depth feedback and guidance.
[‡]Correspondence:{`ferdinand.kapl`,`emmanouil.angelis`,`tobias.hoeppe`}@tum.de.

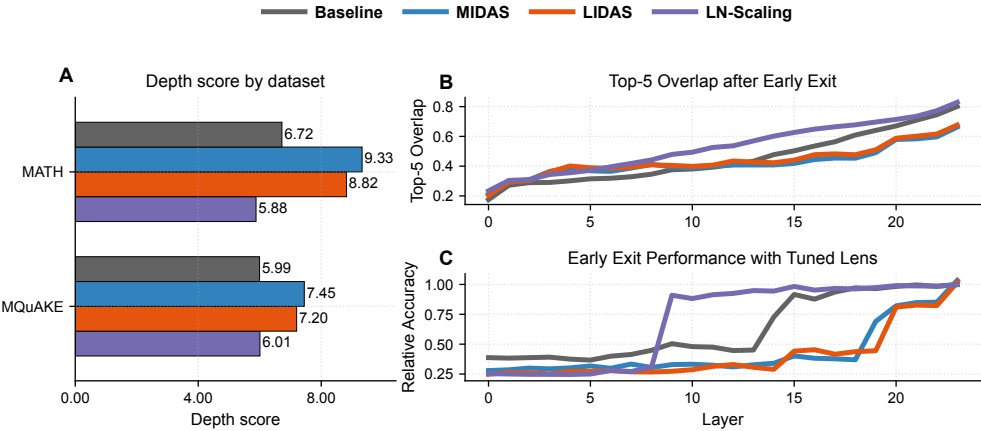

Figure 1: **Depth-grown models use their depth more (1.7B)**. (A) Depth score (Csordás et al., 2025) on MATH (Hendrycks et al., 2021) and MQuAKE (Zhong et al., 2023). (B) Top-5 overlap between each layer's early-exit vocabulary and model's final vocabulary on 20 prompts from GSM8K (Cobbe et al., 2021). (C) Early-exit relative accuracy versus layer on *Variable Assignment Math* reasoning primitive.

Table 1: **Performance comparison of a standard transformer baseline, LayerNorm-Scaling, and the two grown models MIDAS and LIDAS**. Grown models, especially LIDAS, outperform the non-grown baseline on reasoning-heavy tasks such as Math Word and Primitives. LN-Scaling on the other hand, achieves only minor improvements, which diminish when scaling to the larger model.

| | | Standard cooldown | | | | | | | Math cooldown | |
|---|---|---|---|---|---|---|---|---|---|---|
| | | **Holdout Set** (NLL ↓) | **Open-book Q&A** (F1 ↑) | **Closed-book Q&A** (F1 ↑) | **Lambada** (Acc ↑) | **Hellaswag** (Acc ↑) | **Math Word** (Acc ↑) | **Primitives** (Acc ↑) | **Math Word** (Acc ↑) | **Primitives** (Acc ↑) |
| 360M | Baseline | 2.18 | 22.89 | 14.50 | 43.35 | 39.97 | 3.69 | 30.06 | 8.10 | 33.12 |
| | LN-Scaling | **2.16** | 23.14 | **14.89** | 42.17 | 40.00 | 2.89 | **31.38** | 8.45 | 41.26 |
| | MIDAS | 2.18 | 24.57 | 13.75 | 43.31 | 40.36 | **4.39** | 28.18 | **13.43** | 35.14 |
| | LIDAS | **2.16** | **26.63** | 14.57 | **44.03** | **40.58** | 4.36 | 31.20 | 12.30 | **50.36** |
| 1.7B | Baseline | **1.96** | 29.57 | 18.61 | 50.05 | 46.28 | 13.75 | 34.84 | 23.28 | 42.77 |
| | LN-Scaling | 1.97 | 29.11 | 18.63 | 48.94 | 45.44 | 11.00 | 44.38 | 17.84 | 50.58 |
| | MIDAS | 1.97 | 28.80 | 18.50 | 50.81 | 46.19 | 16.07 | 40.88 | 24.01 | **55.46** |
| | LIDAS | **1.96** | **29.84** | **19.08** | **51.41** | **46.32** | **18.59** | **47.34** | **24.60** | 53.00 |

## 2 EVALUATING DEPTH-GROWN TRANSFORMERS: MIDAS & LIDAS

Due to space limitations, we describe the growing procedure as well as the derivation of MIDAS and LIDAS in Appendix A. In the following, we compare both gradual depth-growing methods against the standard non-grown baseline. We additionally compare against LayerNorm-Scaling (LN-Scaling; (Sun et al., 2025)), a non-growing modification of the LayerNorm sublayer proposed to alleviate the *curse of depth* in pre-layernorm Transformers (see Appendix G for details).

**Setup.** We use the 360M and 1.7B models from the SmolLM-v1 family (Ben Allal et al., 2024), trained on 200B and 400B tokens respectively, to probe scaling behavior. All models are trained from scratch on the SmolLM-Corpus, a curated mixture of educational and synthetic texts, as well as mathematics and code. For all grown models we present, we use the block size $b = 4$ and a PROP-1 growing schedule ( Appendix C). We evaluate models on held-out NLL alongside a standard suite of knowledge and reasoning benchmarks. In particular, we include reasoning primitives—synthetic, controlled, multiple choice tasks (e.g., Variable Assignment and Copying variants) designed to isolate multi-step symbolic manipulation and in-context compositional reasoning ( Appendix E).

**Results.** Aggregated results are shown in Table 1. Consistent with the findings of Saunshi et al. (2024), we observe that depth-grown models (MIDAS and even more LIDAS) outperform the baseline and LN-Scaling on reasoning-heavy tasks (i.e. Math Word and Reasoning Primitives). On the

remaining benchmarks (Open-book Q&A, Closed-book Q&A, and Lambada), we observe little deviation from the baseline model with LIDAS being slightly superior to MIDAS. Compute-wise, we observe a 29% speedup in training for depth-grown models against the non-grown variants (Table 7). To stabilize results on Math Word, we additionally report the performance of models which are finetuned on the OpenWebMath dataset. While the relative order stays the same, we observe for the 360M model that the improvements for the grown models become more pronounced. Additionally, we report the performance of the 360M baseline and its growing variants on another seed in Table 8, confirming that the above findings are robust.

## 3 DEPTH ANALYSIS

We now study how gradual depth growth reshapes computation across depth through three targeted questions: (i) whether grown models utilize depth more effectively, (ii) whether interchangeable computational blocks are developed (iii) whether growth induces structured, block-wise layer roles. In Appendix B.2, we address a fourth question: what is the relation of MIDAS with LIDAS regarding weight symmetry and contribution per attention matrix, connecting it to benchmark results of the previous section. For each question, we follow a hypothesis–experiment–evidence procedure using depth-wise analyses and interventions inspired by Csordás et al. (2025). All analyses are conducted on the 1.7B models from Table 1, with corresponding 360M results deferred to Appendix F.2. Experimental details are provided in Appendix D. For notation, we follow Csordás et al. (2025): $h_{i+1}$ denotes the residual stream after transformer layer $l_i$, $a_i$ the layer's attention output and $m_i$ the output of the MLP.

### 3.1 DOES DEPTH GROWTH LEAD TO DIFFERENT DEPTH UTILIZATION?

**Hypothesis.** Gradual depth-grown Transformers (with MIDAS and LIDAS) utilize model depth more efficiently than conventionally trained, non-grown Transformer baselines.

**Evidence.** Skipping late layers degrades prediction accuracy substantially more for MIDAS and LIDAS than for the baseline, which coincides with an increased depth score.

**Experiments.** To investigate the contribution of deeper layers, we evaluate intermediate representations via a Tuned Lens [1] (Belrose et al., 2023). For each layer $l_i$, we train an affine adapter on a split of FineWeb-Edu to map the layer's output to the final hidden state and compute logits via the final lm-head. We then measure depth use via top-5 vocabulary overlap and early-exit accuracy (Figure 1B-C). Finally, we compute the depth score (see Appendix D) to summarize where computation occurs along the network by estimating each layer's influence on future tokens (Figure 1A).

**Interpretation.** For MIDAS and LIDAS, Figure 1B shows that early-exit predictions differ substantially more from the final logits than in the baseline (lower top-5 overlap), indicating that later layers in the grown models add features to the residual stream that are required for the final prediction. In Figure 1C, the baseline reaches its final performance by Layer 18, whereas accuracy for both grown models continues to improve up to the last layer. Lastly, Figure 1A consistently reports higher depth scores for the grown models across datasets, most notably on math tasks, indicating that more computation is concentrated in later layers.

### 3.2 DOES DEPTH GROWTH FORM PERMUTABLE COMPUTATIONAL BLOCKS?

**Hypothesis.** Non-grown models depend on their specific layer ordering. Depth-grown models, on the other hand, develop computational blocks that are robust to block-level ordering interventions.

**Evidence.** Reduced performance degradation under multi-layer perturbations indicates lower layer order dependence and greater robustness of MIDAS and LIDAS.

---

[1] Note that this should result in more accurate predictions than naively applying the unembedding matrix at every layer (LogitLens (Nostalgebraist, 2020)), as done in Csordás et al. (2025).

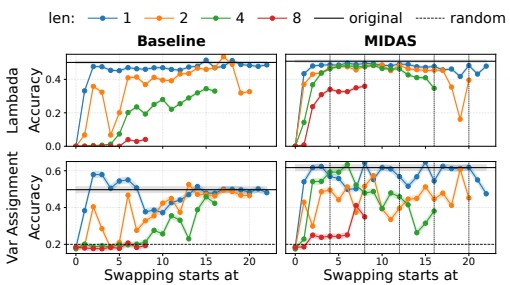 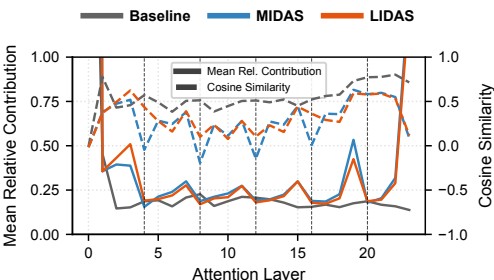

(a) **Effect of swapping blocks of layers on Lambada and reasoning primitive *Variable Assignment Math*.**

(b) **Attention layer contributions to the residual stream and Cosine Similarity.**

**Experiments.** To evaluate layer functional independence, we swap contiguous sub-blocks of sizes $\{1, 2, 4, 8\}$ and measure the effect on downstream performance.

**Interpretation.** Swapping single layers has little effect on either baseline or grown models (except near the input; Lad et al. (2024)). As block size increases, baseline accuracy rapidly deteriorates, whereas grown models tolerate swaps of up to four layers with only minor degradation, indicating reduced order dependence. Even when swapping middle 8-layer blocks, grown models remain above random performance while the baseline collapses. These patterns are consistent with the emergence of computational blocks whose function depends more on the block's presence than on its internal ordering (Figure 2a).

### 3.3 DOES GRADUAL GROWTH FORM LAYER-WISE PATTERNS?

**Hypothesis.** The block-wise growing introduces a cyclical pattern in the architecture such that each layer within a block fulfils a certain role.

**Evidence.** The contribution of the attention sublayer, in norm and cosine similarity, repeats in each block. When performing causal interventions, the effect for each layer within a block also repeats. Reversing the order of layers within and especially across blocks destroys the performance of grown models more than swapping, where local order is more preserved.

**Experiments.** Using the tools of Csordás et al. (2025), we compute for each attention layer its cosine similarity to the residual stream ($\frac{a_i \cdot h_i}{||a_i|| \, ||h_i||}$) and its mean relative contribution $\frac{||a_i||}{||h_i||}$. Moreover, in Appendix B.1, we further conduct intervention experiments that skip individual Transformer (sub)layers or reverse the order of four consecutive layers, providing also the corresponding interpretation.

**Interpretation.** Grown models exhibit a highly cyclical pattern in the middle, where the effect is especially visible for the attention sublayer (Figure 2b). The mean relative contribution of the attention sublayer always grows from its lowest point at the first layer of every block to its highest point at the last layer of the block. The highest spike across depth is always at the final layer of the last block in the middle of the network, i.e., the overall second-to-last block. For MIDAS the cosine similarity of the attention sublayers in the middle, similarly to their contributions, always rises from around zero, adding orthogonal features, or slightly negative, weakening or erasing features, to the highest but only slightly positive cosine similarity at the end of each block. The pattern for LIDAS is a little bit less clear, but the cosine similarity never drops as low as MIDAS, potentially adding features from subspaces that are better aligned with the residual stream across the whole block.

## 4 CONCLUSION

We studied why gradual depth growth improves reasoning in large language models. Depth-grown models outperform conventional baselines on reasoning while reducing training cost. Our analyses suggest that these gains come from better depth utilization. Overall, gradual depth growth offers a practical route to more efficient and capable language models.

## REPRODUCIBILITY STATEMENT

We conduct all training and experiments using publicly available code and datasets. Our training setup builds on the open-source `nanotron`[2] library; Table 2 lists all model and optimizer hyperparameters, and Appendix C provides the exact SmolLM training mixture with links to each public dataset to fully reconstruct the training corpus. Additionally, we provide a detailed description of the growing operators in Appendix A and further detail in Appendix C. To reproduce our analyses, Appendices D and E detail the evaluation protocols and the open-source libraries we use, along with any task-specific settings.

## ACKNOWLEDGMENTS AND DISCLOSURE OF FUNDING

The authors would like to thank João Sacramento for insightful discussions and support throughout this work.

This work was partially supported by the Helmholtz Foundation Model Initiative and the Helmholtz Association. The authors gratefully acknowledge the Gauss Centre for Supercomputing e.V. (www.gauss-centre.eu) for funding this project by providing computing time through the John von Neumann Institute for Computing (NIC) on the GCS Supercomputer JUPITER — JUWELS (Jülich Supercomputing Centre, 2021) at Jülich Supercomputing Centre (JSC). Furthermore, the authors appreciate the computational resources provided by the National High Performance Computing Centre (www.nhr.kit.edu). The research presented is supported by the TUM Georg Nemetschek Institute Artificial Intelligence for the Built World and the German Federal Ministry of Education and Research (Grant:01IS24082).

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

## DISCLOSURE OF LLM USAGE

Large language models were used for language editing, such as enhancing clarity, precision, and flow, and for minor aesthetic adjustments to figures to improve interpretability.

## A    THE GROWING OPERATOR

We fix a base architecture class $\mathcal{F}$ (width, heads, embedding size, etc. are fixed) and vary only depth. Let $f_L \in \mathcal{F}$ denote a model with $L$ Transformer layers, written as an ordered list $f_L = [\ell_0, \ldots, \ell_{L-1}]$. A (depth) *growth operator* $G : \mathcal{F} \times \mathbb{N} \rightarrow \mathcal{F}$ maps an $L$-layer model to an $(L + b)$-layer model, such that $G(f_L; b) = f_{L+b}$, where $b \in \mathbb{N}$ is the *block size* (the number of layers added per growth step). Following Saunshi et al. (2024), we consider growth operators that insert new layers in the centre of the model and keep the block size $b$ fixed across growing stages.

The following strategies use layer duplication to initialize new layers within the newly inserted block. This consists of deep copying all parameters within the layer, including their optimizer state. The result is two initially identical copies at different depths of the model, thus allowed to diverge as training continues.

**MIDAS.** When depth increases by $b$ layers per stage, at stage $n$ we have $L = nb$ and can partition $f_L$ into $n$ contiguous blocks of size $b$ :

$$f_L = [B_0 \,\|\, B_1 \,\|\, \cdots \,\|\, B_{n-1}], \quad B_j = [\ell_{jb}, \ldots, \ell_{(j+1)b-1}]. \tag{1}$$

Let $m_b = \lceil \frac{n}{2} \rceil - 1$ denote the *middle block* index. Middle gradual stacking inserts a new block $B'$ immediately after $B_{m_b}$, i.e.

$$G(f_L; b) = [B_0 \,\|\, \cdots \,\|\, B_{m_b} \,\|\, B' \,\|\, B_{m_b+1} \,\|\, \cdots \,\|\, B_{n-1}]. \tag{2}$$

If $B' = B_{m_b}$ (copying the middle block), we recover MIDAS as proposed in Saunshi et al. (2024).

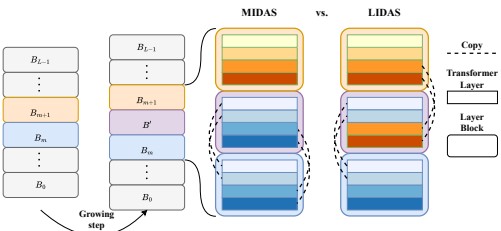

Figure 3: **Illustration of growing strategies with block size 4**: MIDAS vs. LIDAS, with an even number of existing blocks. MIDAS (Saunshi et al., 2024) simply copies $B' = B_m$, which is the block preceding mid-depth. When seen from a block-wise perspective instead of a layer-wise perspective, our proposed variant LIDAS may be interpreted as forming $B'$ from the two blocks surrounding the mid-depth by combining the first two layers of $B_{m+1}$ with the last two layers of $B_m$
. This small difference in initialization leads to significantly improved performance as shown in Table 1.

**LIDAS.** Since we are constrained by the block patterning in MIDAS, we propose Layer-wise mIDdle grAdual Stacking, or LIDAS, in which we consider the *middle layer* $m_l = \lceil \frac{L}{2} \rceil$ to be the central point of the growing operation. We then construct a new block $B' = [l_{m_l-\lceil b/2 \rceil}, \ldots, l_{m_l+\lfloor b/2 \rfloor}]$, around the middle layer $l_{m_l}$, which is inserted after the layer $l_{m_l+\lfloor b/2 \rfloor}$. For an odd number of blocks, MIDAS and LIDAS coincide by selecting the same layers. They differ for an even number of blocks, shown from a block-wise perspective in Figure 3. Further details can be found in Appendix C.

**Training runs and schedules.** A training run is specified by (i) the model class $\mathcal{F}$, (ii) the target depth $L_{\text{final}}$, (iii) the initial depth $L_0$ (typically $L_0 = b$), (iv) a fixed block size $b$, and (v) a stage schedule $\{T_s\}_{s=0}^{S-1}$ (training steps per stage). Starting from $f_{L_0}$, after each stage $s$, we apply $G(\cdot; b)$ to obtain $f_{L_{s+1}}$ with depth $L_{s+1} = L_s + b$. We repeat until $L_{S-1} = L_{\text{final}}$.

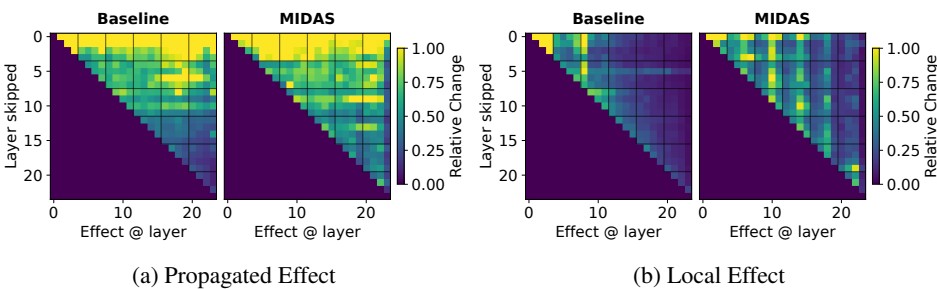

(a) Propagated Effect                    (b) Local Effect

Figure 4: **Baseline vs. `MIDAS`. Effect of skipping a layer on downstream layer contributions for *future* tokens**. (a) `MIDAS` relies more on later layers than the baseline for future computations. Especially skipping the second layer of each mid-block strongly impacts the immediately following layer. (b) For `MIDAS`, the third layer of every block in the middle directly depends on all previous computations. We refer to Figure 11 and Figure 12 for results including `LIDAS`.

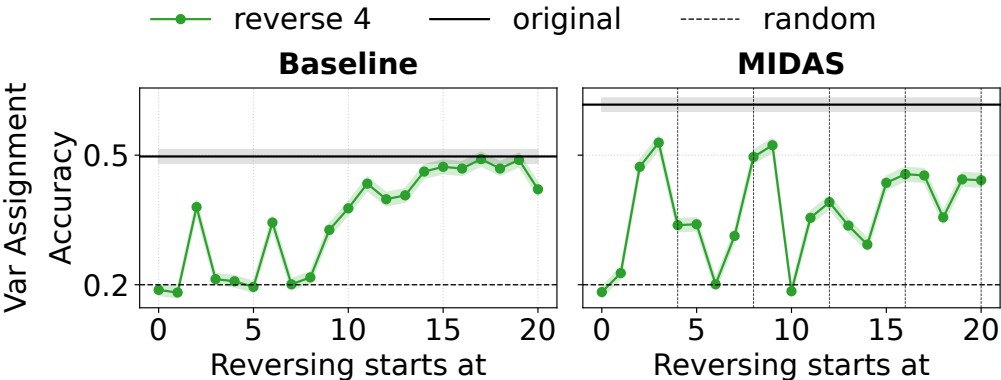

Figure 5: **Effect of reversing the order of four consecutive layers on reasoning primitive.** Reversing the order of layers within a block (first layer of each grown block as vertical grey lines; right figure) of size 4 degrades the performance for grown models more than swapping the same number of layers (len$=2$ in Figure 2a). The baseline is more robust to reversing the order of the later layers, while `MIDAS` is especially sensitive to reversing the order across grown blocks, i.e., the last two and first two layers of consecutive blocks. Starting to reverse at these positions, which correspond to layer index 6, 10, and 14, always results in a substantial drop in performance. Figure 10 shows results including `LIDAS` and an additional dataset.

## B  DEPTH ANALYSIS SUPPLEMENTARY RESULTS

Here we present additional analysis for Section 3. Specifically, we complete the experiments and interpretations related to the periodic-patterns conjecture (Appendix B.1) and, in Appendix B.2, compare `MIDAS` and `LIDAS` with respect to their weight structure and attention behavior.

### B.1  DOES GRADUAL GROWTH FORM LAYER-WISE PATTERNS?

**Experiments Continued.** We intervene by skipping a transformer layer or sublayer and track the relative changes in downstream computations under two regimes: (i) propagated: zeroing that component's contribution to the residual stream and forwarding this change to all downstream layers; and (ii) local: removing a layer's contribution from all subsequent inputs separately to isolate pairwise source–target dependencies. Finally, we assess the effect of reversing the order of four consecutive layers and comparing the outcome to results from Figure 2a. A detailed explanation of the interventions can be found in Appendix D.

**Interpretation.** Turning towards interventions, by skipping a layer, the most pronounced disruption to future computations arises when skipping the second layer of each block (aside from the earliest

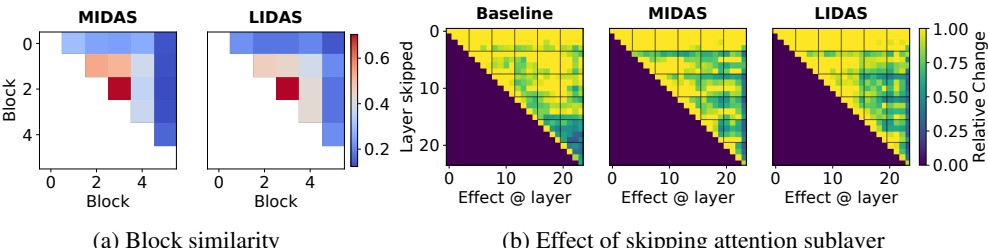

(a) Block similarity          (b) Effect of skipping attention sublayer

Figure 6: **Baseline vs. MIDAS vs. LIDAS.** (a) The weight similarity, measured by cosine similarity, between feedforward layers per block is more symmetric for LIDAS compared to MIDAS. We omit the baseline as its weight similarities are all close to zero. (b) Skipping the first attention sublayer of every block in the centre of the network has a lower effect on the following layers' *current* computations in MIDAS compared to LIDAS.

layers), with often the biggest observed relative change in the immediate layer after it, i.e., in each block's third layer (Figure 4a). We hypothesize that the second layer prepares features for future computations. If we measure the relative change on the following layers directly, we notice a clear and striking pattern (Figure 4b). For future computations, the third layer of every block directly depends on the features of almost all previous layers, potentially performing an aggregating operation. The direct change of removing the output of a previous layer is less on deeper blocks that can depend on more inputs simultaneously, i.e., visually a fading pattern. The last block mostly depends on the final aggregation and strengthening of relevant features performed by the second-to-last block.

Reversing the order of four consecutive layers (Figure 5) reduces performance in the grown model far more than swapping pairs of two or four layers (len $= 2, 4$ in Figure 2a), where local order is more preserved. The baseline is comparatively robust to reversals, which aligns with the hypothesis from (Csordás et al., 2025) that later layers in pre-layernorm transformers refine the current output distribution with less order dependence. By contrast, the grown model is most brittle when the reversal straddles block boundaries, showcasing that the order of layers within a block matters.

## B.2 DOES GROWING STRATEGY LEAD TO DISTINCT BEHAVIOUR?

> **Hypothesis.** Compared to MIDAS, LIDAS produces more symmetric weights and engages the attention sublayers in the central blocks more strongly, which we hypothesize contributes to its better empirical performance.
>
> **Evidence.** In LIDAS, inter-block cosine similarities are higher and more symmetric about the centre. Skipping the first attention sublayer in the middle blocks causes larger relative changes in the hidden state of the token under consideration.

**Experiments.** To measure the weight similarity of blocks for the grown model, we concatenate all weights from the feedforward layers of a block and calculate the cosine similarity to other blocks. Similarly to before, we skip layers and measure the relative change for all later layers, but now on *all* tokens (including the current token).

**Interpretation.** In LIDAS we observe a block-similarity structure that is symmetric about the model's centre, whereas in MIDAS the central block is more similar to the earlier (upper) blocks than to the later (lower) ones, yielding an asymmetric pattern (Figure 6). This difference follows from the growth rule: LIDAS duplicates the exact layer-wise middle, while MIDAS is constrained to the nearest block centre. With an even number of blocks, the MIDAS choice necessarily biases similarity toward one side. Additionally, this growing strategy leads to a higher utilisation of the first attention sublayer of every block (Figure 6b), making it more aligned with the residual stream and having a larger effect on future layers. This effect is especially noticeable for deeper networks (Figure 20), but we also observe it here for the 1.7B model with 24 layers.

## C SMOLLM: ARCHITECTURE & DATA

**Data.** For all SmolLM models we trained, we followed the SmolLM-v1 data mixture from Ben Allal et al. (2024).

- **FineWeb-Edu (deduplicated)** (Ben Allal et al., 2024): Educational slice of FineWeb selected with a Llama3-70B–trained "educational quality" classifier. We use the deduplicated subset ($\approx$220B tokens) included in the SmolLM-Corpus.
- **OpenWebMath** (Paster et al., 2024): High-quality mathematical web pages extracted from Common Crawl with math-aware parsing, quality filtering, and deduplication ($\approx$14.7B tokens). Used to enrich math/reasoning coverage.
- **Cosmopedia v2** (Ben Allal et al., 2024): Synthetic textbooks, stories, and code generated with Mixtral-8×7B using curated topic lists and seed pages. v2 totals $\approx$39M documents ($\approx$28B tokens of textbooks/stories).
- **Python-Edu** (Ben Allal et al., 2024): Educational Python subset built by training an "educational code" classifier on annotated samples from *The Stack* and applying it to the StarCoder training corpus. It contains $\approx$4B tokens with strict quality thresholding.

Given a fixed training-token budget, we then sample the corpus by proportion—70% FineWeb-Edu (deduplicated), 15% Cosmopedia v2, 9% Python-Edu and 6% OpenWebMath. Note that this leads to significant upsampling of the smaller datasets like Python-Edu and OpenWebMath.

**Model architecture.** Both sizes follow a LLaMA-style, decoder-only Transformer with RMSNorm, SwiGLU MLPs, and RoPE positional embeddings (tied input/output embeddings). The 360M variant uses GQA.

|  | SmolLM-1.7B | SmolLM-360M |
|---|---|---|
| Layers | 24 | 32 |
| Model width | 2048 | 960 |
| FFN dimension | 8192 | 2560 |
| Attention heads | 32 | 15 |
| KV heads | 32 (MHA) | 5 (GQA) |
| Norm | RMSNorm | RMSNorm |
| MLP activation | SwiGLU | SwiGLU |
| Batch size | 2M | 1M |
| Learning rate $\eta_{\max}$ | 0.0005 | 0.003 |
| Weight decay | 0.01 | 0.01 |
| Positional embeddings | RoPE ($\theta$=10,000) | RoPE ($\theta$=10,000) |
| Context length (pretrain) | 2048 | 2048 |
| Tokenizer | cosmo2 | cosmo2 |
| Tied embeddings | Yes | Yes |

Table 2: **Hyperparameters for both SmolLM models**

**Training.** We train both sizes for 200k iterations. This corresponds to roughly 200B seen tokens for the 360M model and 400B for the 1.7B model. We use a trapezoidal learning-rate schedule with a linear warmup for the first 2000 steps up to the peak rate $\eta_{\max}$, a constant plateau until step 170000, and a 1-sqrt decay over the final 30000 steps (Hägele et al., 2024). We optimise with AdamW (Loshchilov & Hutter, 2019) and apply global gradient clipping at 1.0 for all runs.

**Training with the Growing operator** For SmolLM with gradual depth growth all training hyperparameters match the baseline in Table 2. We use a fixed block size $b = 4$ and insert a new *middle* block after each stage, instantiating either MIDAS (duplicate the middle stage block) or LIDAS (duplicate the layer-wise middle; see Appendix A), while keeping width and attention heads constant. At every growth step we deep-copy all layer parameters and their optimizer state so duplicated layers start identically (same AdamW moments) and then diverge with continued training; embeddings and the

final head are copied unchanged. The number of growth stages is defined by $k = L_{\text{final}}/b$ and $T$ is the total number of training steps. We allocate per-stage budgets using the PROP-$\alpha$ schedule of Saunshi et al. (2024):

$$T_i \;=\; \frac{i^\alpha}{\sum_{j=1}^k j^\alpha}\, T \quad \text{for } i = 1, \ldots, k\,,$$

and use PROP-1 ($\alpha{=}1$) in our experiments. In practice, we round $T_i$ to integers (largest-remainder to keep $\sum_i T_i = T$) and maintain a *single continuous* learning-rate schedule across stages (no LR reset; the scheduler's global step carries over). We set $T = 170{,}000$ so all models reach their final depth before they enter the cooldown phase.

**Compute requirements.** We trained all models on NVIDIA A100 GPUs (40 GB). The large (static baseline) model ran on 128 GPUs for 4.5 days, and the small (static baseline) model ran on 64 GPUs for 1.5 days.

## D    EVALUATION SETUP FOR ANALYSIS EXPERIMENTS

**Codebases.** Depth analyses and interventions follow the methodology of Csordás et al. (2025), which is extended to block-wise skip/swap operations over consecutive layers (block sizes $\{1, 2, 4, 8\}$) and further extended to permuting consecutive layers in arbitrary order. Tuned Lens experiments follow Belrose et al. (2023).

**Reproducibility defaults.** We adopt the default configuration from the original depth-analysis repository of Csordás et al. (2025) for reproducibility. Specifically, we use the same fixed set of GSM8K prompts/examples for early-exit, skip, swap and relative contribution evaluations, and we keep random seeds, batching, and evaluation hyperparameters at their defaults unless stated otherwise.

**Models and data.** We analyze SmolLM-v1 backbones at 360M and 1.7B parameters (training details in Appendix C) and evaluate on MATH, MQuAKE, and GSM8K as described in the main text. Preprocessing follows Csordás et al. (2025).

**Intervention protocols.** We distinguish *heatmap (relative-change) experiments* from *benchmarked interventions*. Heatmaps quantify relative changes and use **single (sub)layer skipping only**. Benchmarked interventions (accuracy-based) are described separately below. For heatmaps, we evaluate two intervention *modes* and two *measurement axes*, following and extending Csordás et al. (2025):

- **Current vs. future effects.** In the *current* setting, we intervene by erasing the entire (sub)layer contribution *for all tokens* and measure changes on all positions. In the *future* setting, for a chosen boundary token index $t$, we erase the (sub)layer contribution *only for tokens $\leq t$*, leaving tokens $> t$ unchanged at that (sub)layer; we then measure changes *strictly on tokens $> t$*. This design directly tests whether information is transferred to later tokens via attention, ruling out purely pointwise (self-only) computation.

- **Output vs. later-layer effects.** For the *output probability distribution*, we compute the L2 norm difference between the softmaxed logits of the intervened and original forward passes, aggregated over the relevant positions (current or future). For the *later-layer effects*, we compute, for each later layer, the *relative change* in the residual contribution (i.e., the norm of the difference in that layer's residual update divided by the norm of the original residual update), again aggregated over the relevant positions.

Concretely for heatmaps, in the *future* effects evaluation we select multiple boundary indices $t$ and, for each $t$, (i) erase the (sub)layer's contribution only at tokens $\leq t$, (ii) keep its contribution intact at tokens $> t$, and then compare the intervened and original runs on (a) softmaxed output distributions at positions $> t$ and (b) residual contributions of all later layers at positions $> t$. This directly tests whether features are moved forward in time (to future tokens) by attention.

For heatmaps, we also include a **local (direct) effects** variant, which isolates pairwise dependencies between a source layer and a later target layer without allowing effects to *propagate* through multiple subsequent layers. Specifically, for a source layer $s$ and a later layer $\ell > s$, we subtract the stored

contribution of $s$ from the residual fed into $\ell$ and record the relative change at $\ell$; we do *not* roll this modification forward beyond $\ell$. This complements the propagated analyses by revealing direct, non-compounded influences.

Heatmap interventions are performed at the layer or sublayer level and are **strictly single-layer**. The current/future distinction applies *only* to these heatmap experiments. Block-wise operations are used solely in benchmarked interventions (below).

**Aggregation for heatmaps.** For heatmap visualizations of later-layer effects, we aggregate by taking the *maximum* relative change across (i) batch examples, (ii) eligible sequence positions, and (iii) multiple chosen boundaries $t$ in the future setting. Concretely, for current effects, we take the max over all positions; for future effects, we take the max only over positions strictly greater than $t$, and then take the max over all tested $t$ for each example. This yields a single matrix of source-layer by target-layer maxima per model/setting.

**Tuned Lens training and evaluation.** Following Belrose et al. (2023), we train, for each layer, a small affine adapter that maps that layer's residual output to the hidden representation with the same shape that serves as the input to the *final* normalization layer immediately before the unembedding. Final logits are then obtained by applying the model's final normalization and unembedding as usual. Adapters are trained on a held-out split of FineWeb-Edu and evaluated by (a) KL divergence between early-exit and final distributions and (b) top-5 vocabulary overlap with the final prediction (cf. Figure 1B/C for 1.7B and Figure 13B/C 360M).

**Benchmarked interventions.** We evaluate accuracy on downstream benchmarks under: (i) Tuned Lens early-exit (using the adapter path described above), (ii) skip interventions, and (iii) swap interventions. For benchmarks, we may intervene on contiguous **blocks** of sizes $\{2, 4, 8\}$ (in addition to single layers). We decode with greedy top-1 and compute benchmark accuracy (e.g., Math Word, reasoning primitives), matching the evaluation protocol used for the unmodified model. The current/future distinction does *not* apply to benchmark evaluations.

**Depth score.** We report the *logit-effect* depth score based on `mean_dout`. For each layer $\ell$, `mean_dout` is the across-examples mean of the maximum L2 change in the softmaxed output distribution at future tokens when intervening at layer $\ell$ (future-setting; see intervention protocols). We normalize this per-layer vector to a probability distribution over layers and take its expected layer index as the depth score (Csordás et al., 2025).

## E   DETAILED BENCHMARK RESULTS

**Setup** In Section 2 we presented aggregated results over several Benchmarks. In this section, we show the detailed results for all models and for completeness, we also report results with `LN-Scaling` and further experiment with the combination of `LN-Scaling` and `LIDAS`/ `MIDAS`. We evaluated all models on these benchmarks using the language model evaluation harness library (Gao et al., 2024).

**Benchmarks** We report negative log-likelihood (NLL) on a held-out validation set from the SmolLM-Corpus. In addition, we follow the knowledge and reasoning benchmarking suite of Saunshi et al. (2024).

Knowledge benchmarks are divided into: (i) Open-book Q&A with provided context (TyDiQA-GoldP, SQuADv2, DROP, QuAC, CoQA), and (ii) Closed-book Q&A without context (TriviaQA, TyDiQA-NoContext, NaturalQuestions, WebQuestions), all evaluated zero-shot. We additionally include Lambada (Paperno et al., 2016) and HellaSwag (Zellers et al., 2019) in their standard settings.

For reasoning, we report aggregated performance on Math Word problems (SVAMP (Patel et al., 2021), ASDiv (Miao et al., 2020), MAWPS (Koncel-Kedziorski et al., 2016)) and the synthetic reasoning primitives of Saunshi et al. (2024), evaluated with five-shot prompting.

**Reasoning Primitives** We implemented the **Reasoning Primitives** following the task descriptions in Saunshi et al. (2024). Induction copying is generated by sampling a sequence of random 3-letter

words (e.g., length 10), selecting a contiguous subsequence (e.g., length 5) from within it, appending that subsequence, and asking for the next token in the original sequence. Variable assignment is generated by sampling variable–value statements and querying a single variable's value. We use the same basic, math and code prompt templates.

An example for a *Copying random words* task would be:

*Prompt*:

```
Fill in the blank:
jic dqy sof uzg ewr oxw osp tkj rvw mnu jic dqy sof uzg ewr ___. ->
```

*Answer*:

```
oxw
```

For an example of *variable assignment* task:

*Prompt*:

```
Fill in blank:

o=23
k=3
t=13
a=1
e=9
o=___. ->
```

*Answer:*

```
23
```

Notice that for the above tasks multiple choice format is used and a 5-shot evaluation setting. This means that the random guessing baseline score is 10% for the Copying task and 20% for the variable assignment task.

|       |                    | CoQA  | DROP  | QuAC  | SquadV2 | TyDi QA (wc) | Mean  |
|-------|--------------------|-------|-------|-------|---------|--------------|-------|
| 360M  | Baseline           | 46.08 | 12.48 | 14.27 | 24.35   | 17.25        | 22.89 |
|       | MIDAS              | 50.00 | 12.75 | 14.10 | 24.96   | 21.06        | 24.57 |
|       | LIDAS              | 51.50 | **15.25** | **15.79** | 28.11 | **22.50**    | **26.63** |
|       | LN-Scaling         | 44.80 | 13.18 | 12.45 | 24.11   | 21.14        | 23.14 |
|       | LN-Scaling + MIDAS | 45.70 | 13.14 | 13.86 | 25.11   | 12.39        | 22.04 |
|       | LN-Scaling + LIDAS | **53.17** | 13.03 | 14.59 | **30.69** | 16.19      | 25.5  |
| 1.7B  | Baseline           | 58.36 | 16.52 | 15.91 | 33.88   | 23.17        | 29.57 |
|       | MIDAS              | 59.35 | 16.88 | 17.30 | 36.06   | 14.39        | 28.80 |
|       | LIDAS              | **63.41** | 17.66 | **17.91** | **36.56** | 13.65    | **29.84** |
|       | LN-Scaling         | 54.94 | **17.84** | 16.61 | 32.78 | **23.37**    | 29.11 |
|       | LN-Scaling + LIDAS | 62.36 | 16.09 | 17.74 | 34.98   | 9.05         | 28.04 |

Table 3: **Open-book QA Benchmarks**
.

**Results** In Tables 3 and 4 we report per-dataset results for Open-Book and Closed-Book QA. In line with Saunshi et al. (2024), both grown models (MIDAS and LIDAS) yield larger gains on Open-Book QA than on Closed-Book QA. Notably, LIDAS 1.7B improves over the 1.7B baseline even on most Closed-Book datasets and remains competitive on the rest, which differs from observations made for MIDAS in Saunshi et al. (2024). Overall, the grown variants confer modest but consistent Open-book gains, whereas LN-Scaling alone yields only small changes relative to the non-grown baseline.

Combining growing with `LN-Scaling` can sometimes improve over the standard `LN-Scaling` setting, but often still falls short when compared to `LIDAS`.

| | | Trivia QA | Web Questions | TyDi QA (nc) | Natural Questions | Mean |
|---|---|---|---|---|---|---|
| 360M | Baseline | 19.23 | **16.78** | 12.98 | 9.01 | 14.50 |
| | MIDAS | 18.90 | 14.58 | 12.64 | 8.89 | 13.75 |
| | LIDAS | **20.80** | 15.61 | 12.14 | 9.73 | 14.57 |
| | LN-Scaling | 20.52 | 16.73 | 12.36 | **9.94** | **14.89** |
| | LN-Scaling + MIDAS | 17.76 | 16.60 | 12.66 | 9.18 | 14.05 |
| | LN-Scaling + LIDAS | 18.23 | 15.32 | **13.67** | 9.26 | 14.12 |
| 1.7B | Baseline | 27.72 | 19.20 | 15.34 | 12.18 | 18.61 |
| | MIDAS | **27.98** | 17.96 | 16.16 | 11.91 | 18.50 |
| | LIDAS | 26.85 | 20.24 | **16.34** | **12.90** | **19.08** |
| | LN-Scaling | 25.12 | **20.89** | 15.71 | 12.79 | 18.63 |
| | LN-Scaling + LIDAS | 25.88 | 20.15 | 15.78 | 11.88 | 18.42 |

Table 4: **Closed-book QA Benchmarks**.

| | | ASDiv | MAWPS Add/Sub | MAWPS Multi-Arith | MAWPS Single-Op | MAWPS Single-Eq | SVAMP | Mean |
|---|---|---|---|---|---|---|---|---|
| 360M | Baseline | 3.34 | 3.67 | **1.72** | 5.66 | 2.75 | 5.02 | 3.69 |
| | MIDAS | 3.77 | 3.67 | 1.15 | 6.29 | 6.42 | 5.02 | **4.39** |
| | LIDAS | **4.64** | 1.83 | **1.72** | **7.55** | 6.42 | 4.01 | 4.36 |
| | LN-Scaling | 2.95 | 0.00 | 1.15 | 4.40 | 5.50 | 3.34 | 2.89 |
| | LN-Scaling + MIDAS | 3.56 | **5.50** | 1.15 | 1.89 | 6.42 | **5.69** | 4.04 |
| | LN-Scaling + LIDAS | 3.21 | 3.67 | **1.72** | 4.40 | **7.34** | 3.68 | 4.00 |
| 1.7B | Baseline | 11.15 | 14.68 | 1.15 | 25.16 | **22.02** | 8.36 | 13.75 |
| | MIDAS | 12.93 | 18.35 | 2.30 | 33.96 | 20.18 | 8.70 | 16.07 |
| | LIDAS | **14.88** | **25.69** | 2.87 | **38.36** | 18.35 | 11.37 | **18.59** |
| | LN-Scaling | 10.20 | 13.76 | 1.72 | 17.61 | 14.68 | 8.03 | 11.00 |
| | LN-Scaling + LIDAS | 14.19 | 22.02 | **3.45** | 31.45 | 21.10 | **11.71** | 17.32 |

Table 5: **Math Word**.

On the reasoning benchmarks, Math Word (Table 5) and Reasoning Primitives (Table 6), improvements at 360M are modest on average, while at 1.7B they become more pronounced. For Math Word, `LIDAS` 1.7B attains the best score on five out of six benchmarks (the exception is MAWPS Single-Equation). For Reasoning Primitives, both `MIDAS` and `LIDAS` surpass the baseline, with `LIDAS` 1.7B leading on copying and on the code/math variable-assignment formats, while `MIDAS` slightly edges `LIDAS` on the basic variable-assignment format. We notice however, that the variance of performance between tasks is much higher compared to language tasks.

| | | Copying (random words) | Copying (real words) | Variable assignment (basic) | Variable assignment (code) | Variable assignment (math) | Mean |
|---|---|---|---|---|---|---|---|
| 360M | Baseline | 15.50 | 13.30 | 20.50 | 58.30 | 42.70 | 30.06 |
| | MIDAS | 13.80 | 14.10 | 20.00 | 52.90 | 40.10 | 28.18 |
| | LIDAS | 14.20 | 19.70 | 24.30 | 51.80 | 46.00 | 31.20 |
| | LN-Scaling | 17.90 | 16.40 | 22.70 | 49.10 | 50.80 | 31.38 |
| | LN-Scaling + MIDAS | 15.80 | 16.00 | **26.30** | **58.70** | 48.40 | 33.04 |
| | LN-Scaling + LIDAS | 21.90 | 19.80 | 25.50 | 57.20 | **52.10** | **35.30** |
| 1.7B | Baseline | 16.80 | 23.60 | 20.80 | 64.20 | 48.80 | 34.84 |
| | MIDAS | 19.30 | 24.60 | **37.00** | 61.50 | 62.00 | 40.88 |
| | LIDAS | **28.40** | **31.00** | 36.70 | 71.80 | 68.80 | **47.34** |
| | LN-Scaling | 24.40 | 26.60 | 23.80 | **75.20** | **71.90** | 44.38 |
| | LN-Scaling + LIDAS | 17.00 | 23.00 | 34.80 | 71.20 | 70.60 | 43.32 |

Table 6: **Reasoning Primitives**.

| Model | PetaFLOPs | Ratio |
|---|---|---|
| 360M Standard | 613527.488 | 1.289 |
| 360M Grown | 476147.897 | 1.000 |
| 1700M Standard | 4813222.102 | 1.288 |
| 1700M Grown | 3736608.182 | 1.000 |

Table 7: **PetaFLOPs used for training 200k iterations with block size of 4 and PROP-1 growing schedule.**

Consistent with our depth analyses, these benchmark trends coincide with higher depth scores and later-layer reliance for grown models, whereas LN-Scaling in our setup does not increase depth utilization relative to the baseline nor improve performance.

In addition to improved reasoning performance, models trained with gradual stacking also require fewer computational resources, as reported in previous works. Specifically, MIDAS and LIDAS only need $\approx 77\%$ of the FLOPs used to train the baseline in Table 7.

# F  ADDITIONAL RESULTS ON DEPTH ANALYSIS

In Section 3 we have shown how growing can alter the structure within the model, leading to better depth utilisation and different altering of the residual stream and robustness. However, these results have mainly been presented for MIDAS on the 1.7B model scale. In this section we want to first add results for LIDAS, to show that it does exhibit the same patterns as MIDAS and also show results on the smaller 360M scale. Finally, we include additional ablation plots for block sizes different from 4.

## F.1  1.7B MODELS

This section extends the main analyses for the 1.7B models to LIDAS. For each hypothesis made in Section 3, we show that our results also hold for LIDAS by resuming each experiment.

**Additional results for Section 3.1**  Figure 7 reproduces the early-exit analysis at 1.7B on Lambada and Variable Assignment, showing that grown models' early exit performance is relatively poor over the entire stack while the baseline saturates much earlier. Notably, this result is stable across tasks with different absolute accuracies, suggesting that reliance on later layers reflects a training-induced computational pattern rather than task difficulty. This complements the Section 3.1 diagnostics and reinforces that depth growth yields genuinely deeper computation at scale.

**Additional results for Section 3.2**  Figure 8 extends the swap-ablation result (Figure 2a) to include LIDAS at 1.7B. Grown models are markedly more robust than the baseline when swapping multi-layer blocks (sizes 2–8), especially in the middle of the network, conforming to the signature of block-level permutability argued in Section 3.2. Figure 9 provides the complementary experiment, in which we skip consecutive layers of different sizes. Together, these interventions support our hypothesis that depth growth organizes computation into mid-network blocks whose presence is crucial but whose internal order is comparatively flexible.

**Additional results for Section 3.3**  Figure Figure 10 shows that grown models are particularly fragile when reversing four-layer windows that pass block boundaries, degrading more than under swapping or skipping, indicating that while blocks are permutable as units, the intra-block progression encodes roles that do not commute, as argued in Section 3.3. We can clearly see that LIDAS also follows this pattern and that it is independent of the task being considered. Figures 11 and 12 reveal repeating mid-block motifs and stronger downstream propagation from later layers in grown models, generalising the results from Figure 4 to include LIDAS at 1.7B. Collectively, these results support that our claims from Section 3.3 do hold for LIDAS and are not task-specific.

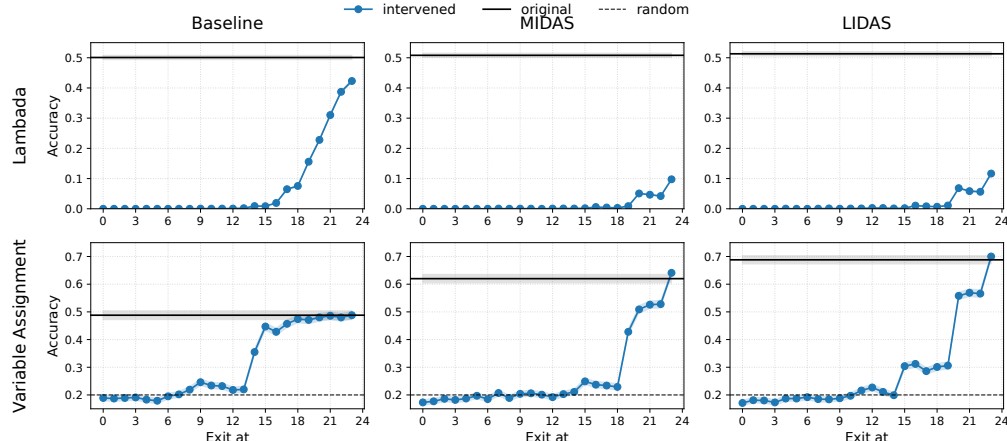

Figure 7: Early exit with tuned lens on *Lambada* and *Variable Assignment Math* for the Baseline, `MIDAS`, and `LIDAS` models at scale 1.7B.

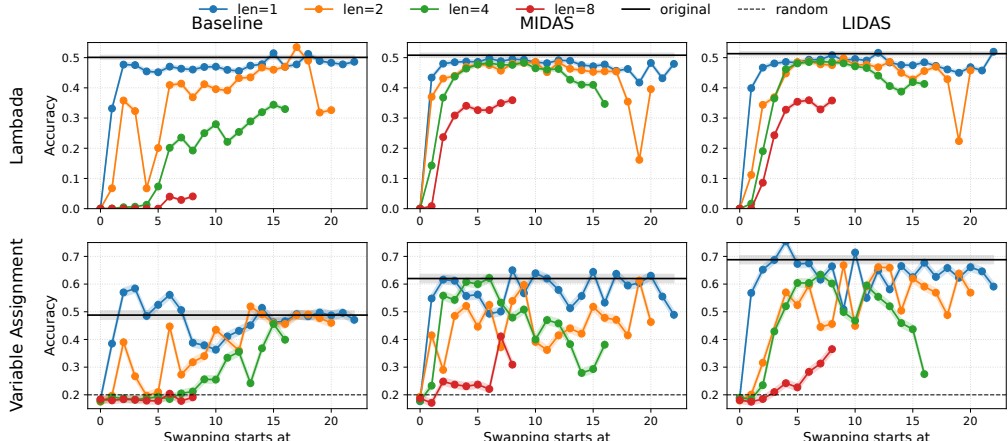

Figure 8: Swap ablations on *Lambada* and *Variable Assignment Math* for the Baseline, `MIDAS`, and `LIDAS` models at scale 1.7B.

### F.2 360M MODELS

This section presents the depth analysis results for the 360M models for the experiments described in the main paper.

**Benchmark results on different seed**   To test the robustness of our benchmark results to random initialisation and data ordering, we retrain the baseline 360M model and grown variants with a different model initialisation and data seed, while keeping all other hyperparameters fixed. As shown in Table 8, absolute scores change only slightly compared to Table 1 and the relative ranking of methods is preserved: both MIDAS and LIDAS match the baseline on language-modelling and knowledge benchmarks, and LIDAS continues to achieve the strongest performance on Math Word and Reasoning Primitives. This indicates that our main conclusions about the benefits of gradual depth growth are stable across seeds.

**Additional results for Section 3.1**   In Figure 13, we summarize 360M model depth utilization using the depth score and tuned-lens early-exit diagnostics (see Figure 1 for the 1.7B case). In Figure 13 A, we observe that the results are more task dependent compared to the 1.7B model: for

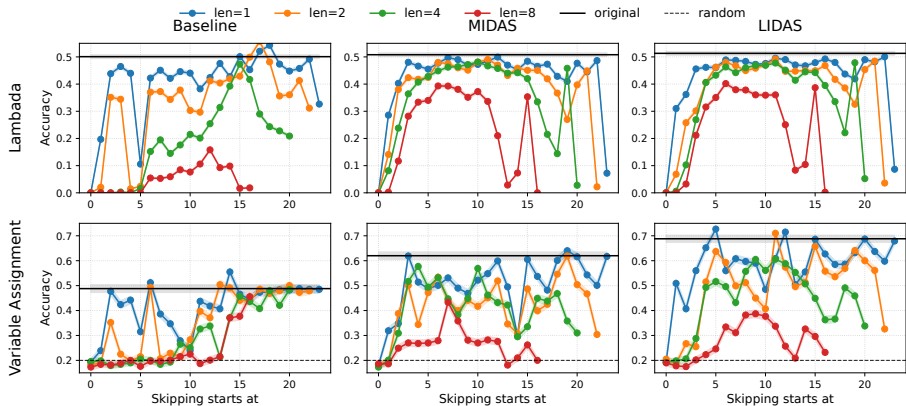

Figure 9: Skip ablations on *Lambada* and *Variable Assignment Math* for the Baseline, `MIDAS`, and `LIDAS` models at scale 1.7B.

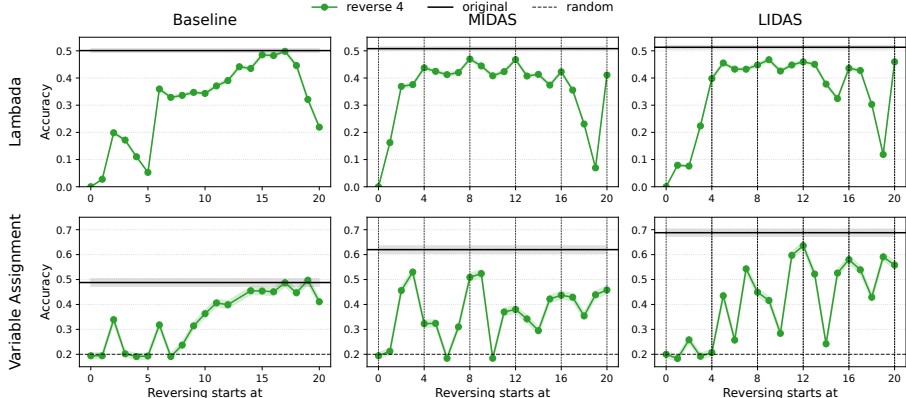

Figure 10: Reversing the order of 4 consecutive layers on *Lambada* and *Variable Assignment Math* for the Baseline, `MIDAS`, and `LIDAS` models at scale 1.7B.

experiments conducted on the MATH dataset, the trend of grown models to utilize more depth is evident (although to a lesser extent compared to the 1.7B model). However, for the MQuAKE dataset, the depth utilization pattern appears more complex, as only `MIDAS` has a higher depth score than the baseline, and no clear conclusion can be derived. In Figure 13 B, early exiting for the baseline model saturates much earlier, consistent with the 1.7B case. Moreover, in Figure 13 C we can also see that peak performance is reached earlier for baseline compared to `MIDAS` and `LIDAS`.

In Figure 14 we replicate benchmarked early exit interventions at 360M: again the picture is less clear compared to the 1.7B model case and the results are task dependent; in Lambada early exiting performance for grown models stays close to 0 until the very end, indicating the necessity of later layers in information processing. In Variable Assignment, however, `MIDAS` and `LIDAS` exhibit different behaviour when compared to the baseline.

**Additional results for Section 3.2** In Figures 15 to 17 we assess robustness under reduced capacity, covering swap, skip, and reversal interventions. Consistent with the 1.7B case, the swap experiments show higher robustness w.r.t block level ordering interventions for the grown models. Moreover, when it comes to the reverse ordering interventions, we observe again this increased sensitivity of the grown models wrt block boundaries.

**Additional results for Section 3.3** We include the small-model counterparts of the future propagated, future local, and current attention ablations (see Figures 18 to 20). In a nutshell, all cyclical patterns observed for the 1.7B model in the main paper still hold for the 360M case: the sensitivity of

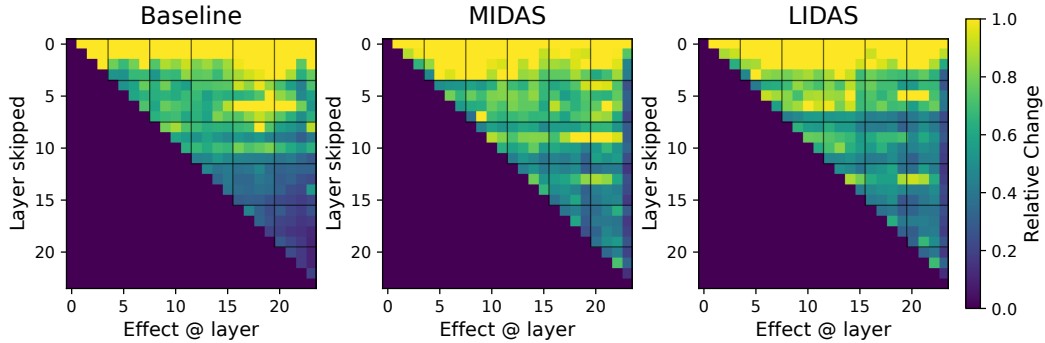

Figure 11: Propagated future effects of single-layer skipping for the Baseline, MIDAS, and LIDAS models at scale 1.7B.

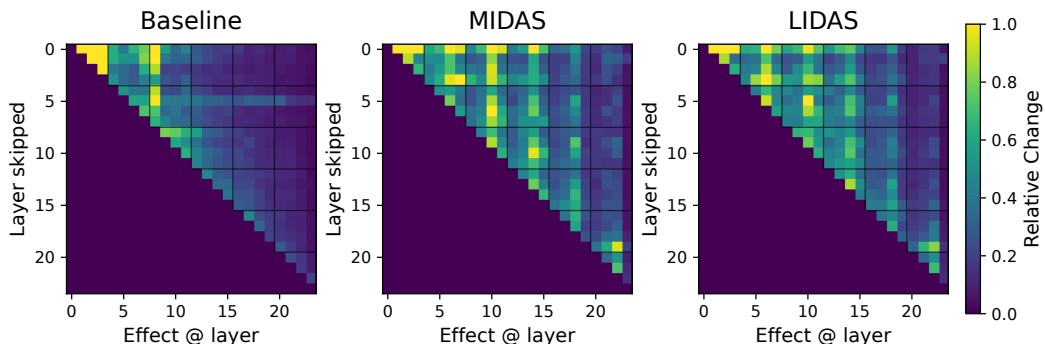

Figure 12: Local future effects of single-layer skipping for the Baseline, MIDAS, and LIDAS models at scale 1.7B.

| | | | Standard cooldown | | | | | | Math cooldown | |
|---|---|---|---|---|---|---|---|---|---|---|
| | | **Holdout Set** (NLL ↓) | **Open-book Q&A** (F1 ↑) | **Closed-book Q&A** (F1 ↑) | **Lambada** (Acc ↑) | **Hellaswag** (Acc ↑) | **Math Word** (Acc ↑) | **Primitives** (Acc ↑) | **Math Word** (Acc ↑) | **Primitives** (Acc ↑) |
| 360M | Baseline | 2.18 | 23.18 | 14.22 | 43.16 | 40.16 | 3.11 | 29.92 | 7.91 | 37.36 |
| | MIDAS | 2.18 | 24.26 | **14.34** | 42.58 | 40.11 | **3.47** | 34.08 | 8.50 | 41.86 |
| | LIDAS | **2.16** | **25.02** | 14.08 | **44.27** | **40.90** | 2.59 | **37.14** | **10.47** | **46.88** |

Table 8: **Performance of the 360M baseline and depth-grown models under a different seed.** We keep the data mixture and training hyperparameters fixed and only change the seed of the random initialisation of model parameters and data order. The results closely match those in Table 1, confirming that the gains of MIDAS and especially LIDAS on reasoning benchmarks are robust to the choice of seeds.

the 3rd layer of each block to the output of all its previous layers Figures 18 and 19 and also reduced impact of the first attention sublayer of every block to later layers for MIDAS model. Notice that these effects are even more pronounced for the 360M model compared to the 1.7B case (compare the corresponding light and dark stripes in Figures 19 and 20 to these in Figures 6b and 12)

For completeness, we also show block-similarity structure at 360M (Figure 21), which mirrors the symmetry patterns observed at 1.7B (cf. Figure 6a).

## F.3 ABLATING BLOCK SIZE

We report mean relative contribution and cosine similarity plots for two ablated model settings: 360M models with block size 8 (Figure 22) and 1.7B models with block size 3 (Figure 23). As in Figure 2b, we observe clear patterns throughout block computation: for the 1.7B MIDAS and LIDAS models,

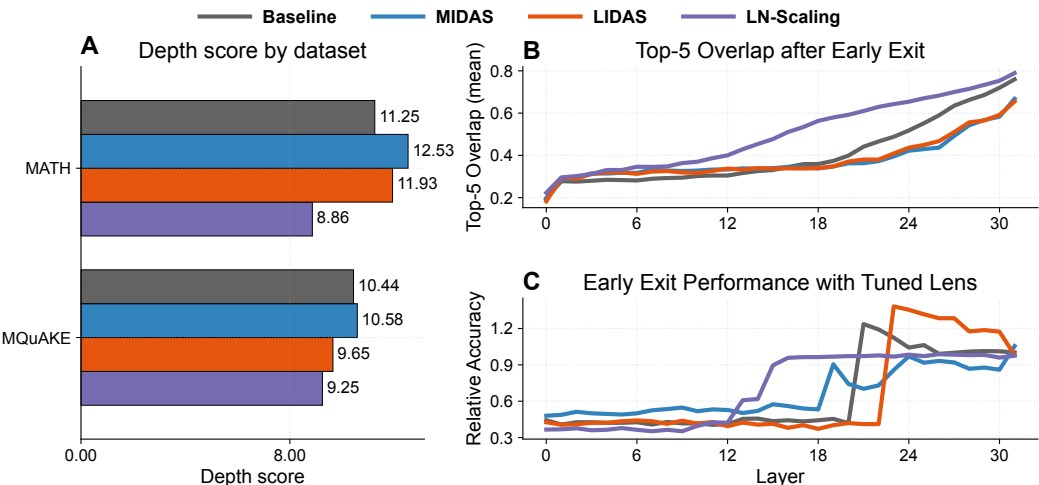

Figure 13: **Depth-grown models use their depth more (360M)**. (A) Depth score (Csordás et al., 2025) on MATH (Hendrycks et al., 2021) and MQuAKE (Zhong et al., 2023). Grown models (MIDAS, LIDAS) have consistently higher depth scores, except LIDAS on MQuAKE. (B) Top-5 overlap between each layer's early-exit vocabulary and model's final vocabulary on 20 prompts from GSM8K (Cobbe et al., 2021). Both grown models studied in this work exhibit lower overlap at later layers, indicating that these later layers still contribute additional features necessary for the final prediction. (C) Early-exit relative accuracy versus layer on *Variable Assignment Math* reasoning primitive. The baseline reaches its best performance early, whereas accuracy for MIDAS and LIDAS is the highest at later layers. Using these metrics, however, LN-Scaling shows no discernible benefit over the baseline in depth utilisation.

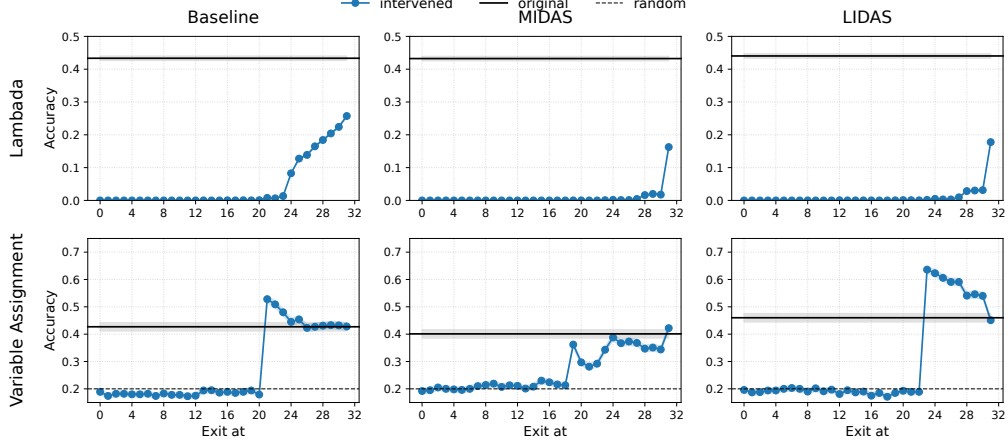

Figure 14: Small models (360M): early exit with tuned lens on *Lambada* and *Variable Assignment Math* for Baseline, MIDAS, and LIDAS.

both mean relative contribution and cosine similarity peak at the last layer of every block, and in the 360M grown models cosine similarity is likewise maximized at the final layer of each block. These experiments suggest that the emergence of such patterns is a characteristic of the growing training method itself, rather than a peculiarity of a specific block size.

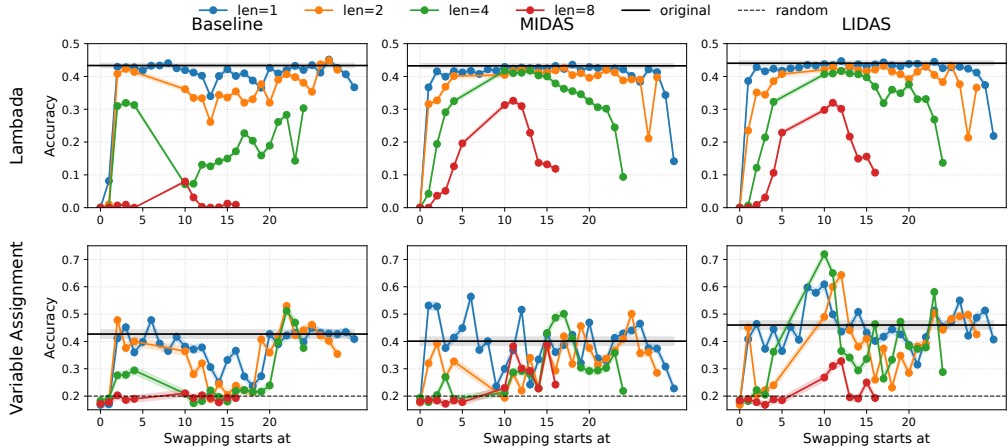

Figure 15: Small models (360M): swap ablations on *Lambada* and *Variable Assignment Math*.

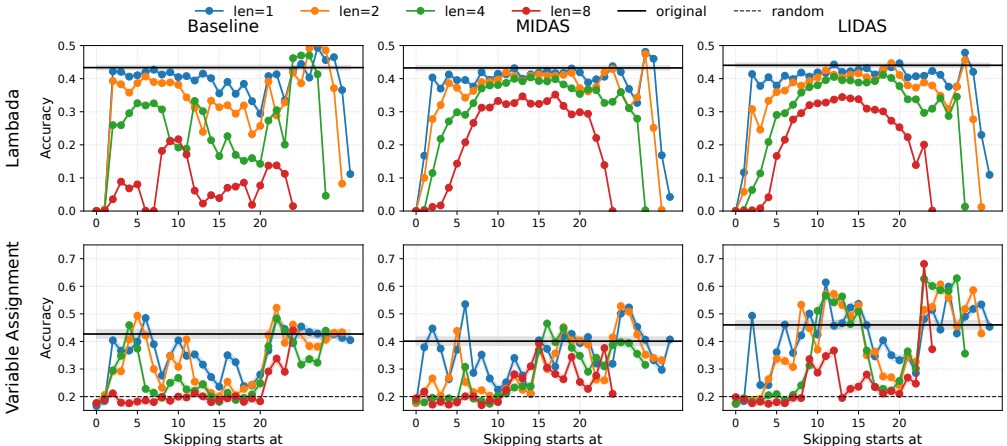

Figure 16: Small models (360M): skip ablations on *Lambada* and *Variable Assignment Math*.

## G    LAYER NORM SCALING

LayerNorm-Scaling (`LN-Scaling` Sun et al. (2025)) is a method that modifies the layer norm sublayer of pre-layernorm transformer architectures with the purpose of increasing the depth usage of later layers. It scales (the variance of) the output $h'_l$ of the layer normalization inversely by the square root of its depth

$$h'_l = \frac{1}{\sqrt{l}}\text{LayerNorm}(h_l)$$

where $h_l$ is the input of the layer norm sublayer. This simple modification mitigates the output variance explosion of deeper Transformer layers, improving their contribution. Additionally, it preserves the training stability common to all pre-layernorm models, which is demonstrated both theoretically and experimentally. Following the recommendation in Sun et al. (2025), we adopt the unscaled initialization strategy, i.e., no additional parameter scaling at initialization based on network depth.

In our setting, however, `LN-Scaling` does not improve depth utilization according to the three diagnostics we used in Figure 1. At both scales, the depth score shifts earlier, top-5 early-exit overlap

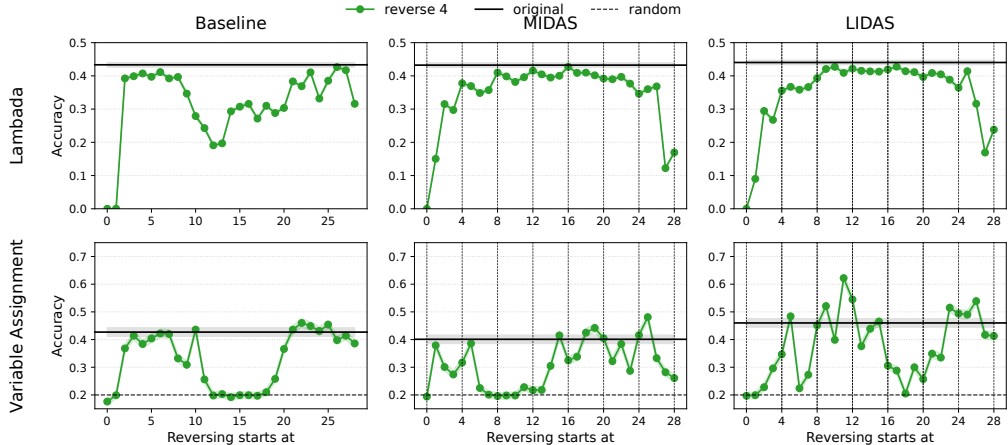

Figure 17: Small models (360M): reversing the order of 4 consecutive layers on *Lambada* and *Variable Assignment Math*.

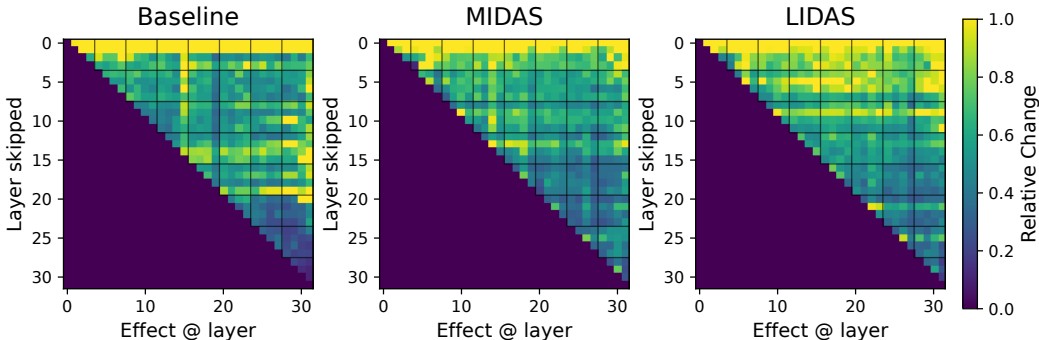

Figure 18: Small models (360M): propagated future effects of single-layer skipping.

increases at earlier layers, and tuned-lens early-exit accuracy plateaus sooner than for the baseline and grown models (Figures 1 and 13).

To probe this further, we evaluate *future-propagated*, *future-local*, and *current-attention* ablations under LN-Scaling (see Appendix D for definitions of these interventions). Across both 1.7B (Figure 24) and 360M (Figure 25) models, interventions to later layers produce smaller downstream effects than in the Baseline, LIDAS, and MIDAS models, supporting our earlier finding that LN-Scaling concentrates computation earlier rather than improving later-layer usage. In line with Table 1, the apparent effectiveness of LN-Scaling diminishes at larger scales. This scale sensitivity may explain the discrepancy with Sun et al. (2025), which does not scale to larger settings.

To understand if MIDAS and LIDAS are also effective for this new architecture, we investigated the depth utilization when combining LN-Scaling and growing. In Figures 26 and 27 we find, that growing can also increase the depth usage of architectures using LN-Scaling, indicating the generality of our findings.

Furthermore, we find that LN-Scaling does not yield substantial gains over the baseline and is typically outperformed by LIDAS, particularly at the 1.7B model size. When combining LN-Scaling with LIDAS Table 9, we observe consistent improvements over the LayerNorm-scaled baseline for the 360M model in all categories except Closed-book Q&A, and in reasoning-heavy tasks also for the 1.7B model.

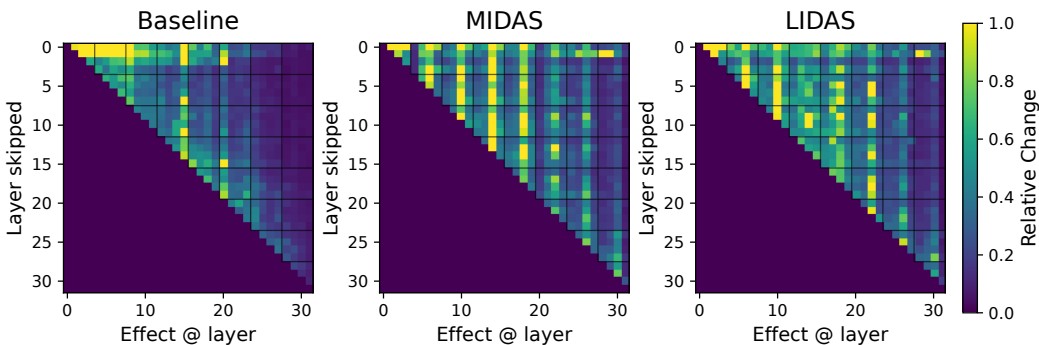

Figure 19: Small models (360M): local future effects of single-layer skipping.

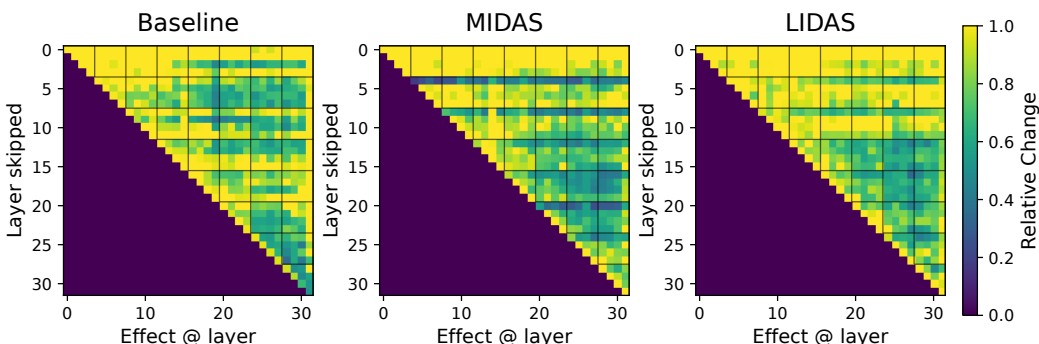

Figure 20: Small models (360M): current effects when skipping the attention sublayer.

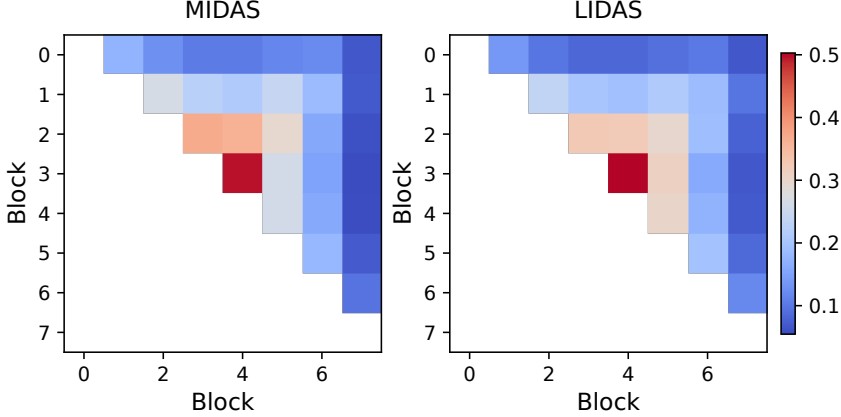

Figure 21: Small models (360M): block similarity for MIDAS and LIDAS.

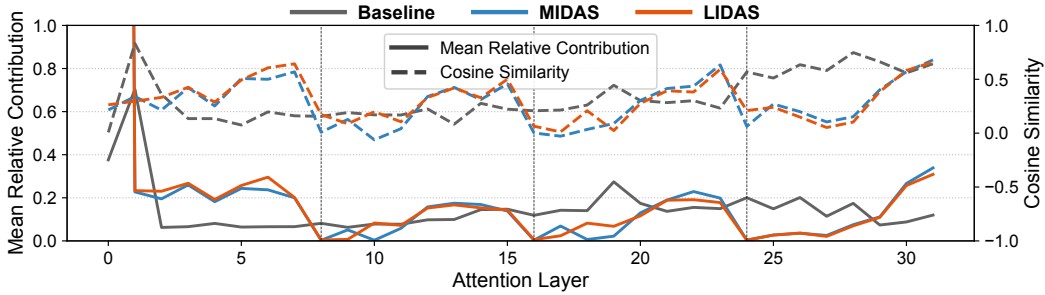

Figure 22: 360M models with block size 8: Mean Relative Contribution and Cosine Similarity plots.

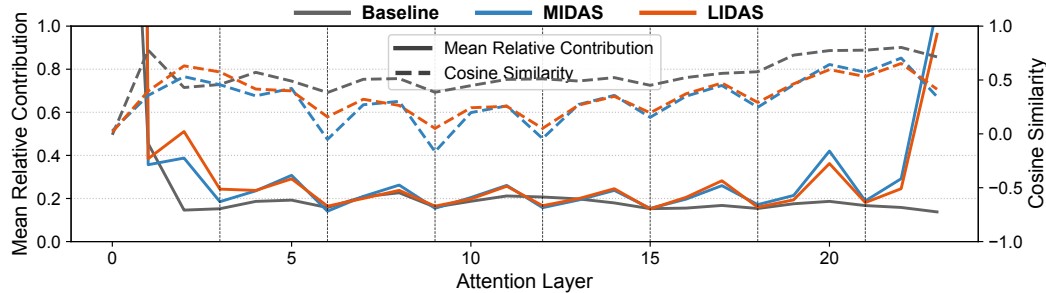

Figure 23: 1.7B models with block size 3: Mean Relative Contribution and Cosine Similarity plots.

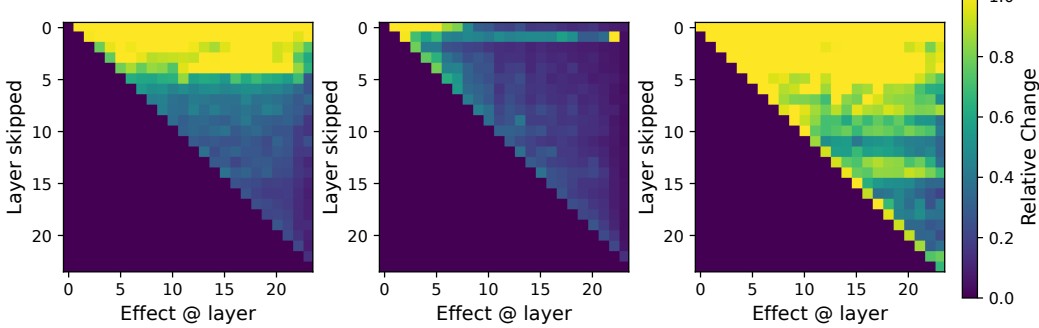

Figure 24: Baseline (1.7 B) model combined with `LN-Scaling`: (from left to right) future propagated layer, future local layer and current attention effects heatmaps

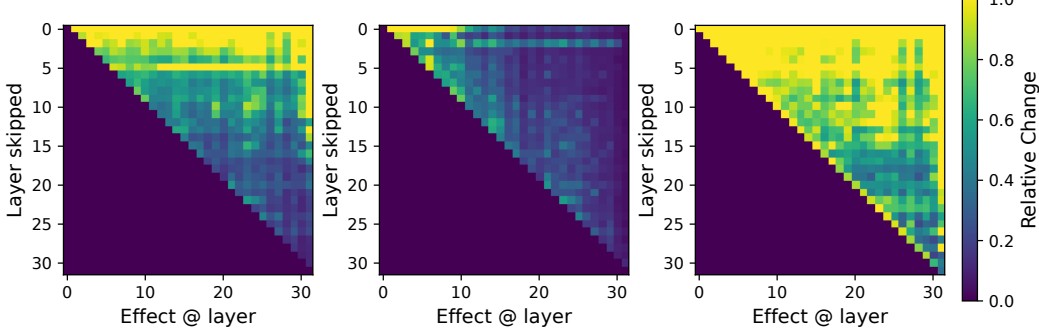

Figure 25: Baseline (360 M) model combined with `LN-Scaling`: (from left to right) future propagated layer, future local layer and current attention effects heatmaps

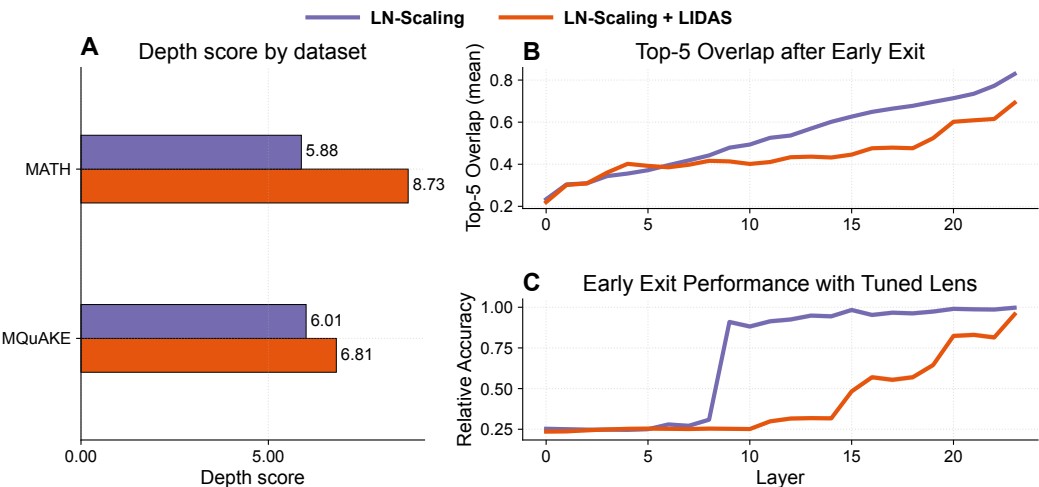

Figure 26: **Depth-grown models use their depth more also when applying growing techniques to `LN-Scaling` (1.7B)**.

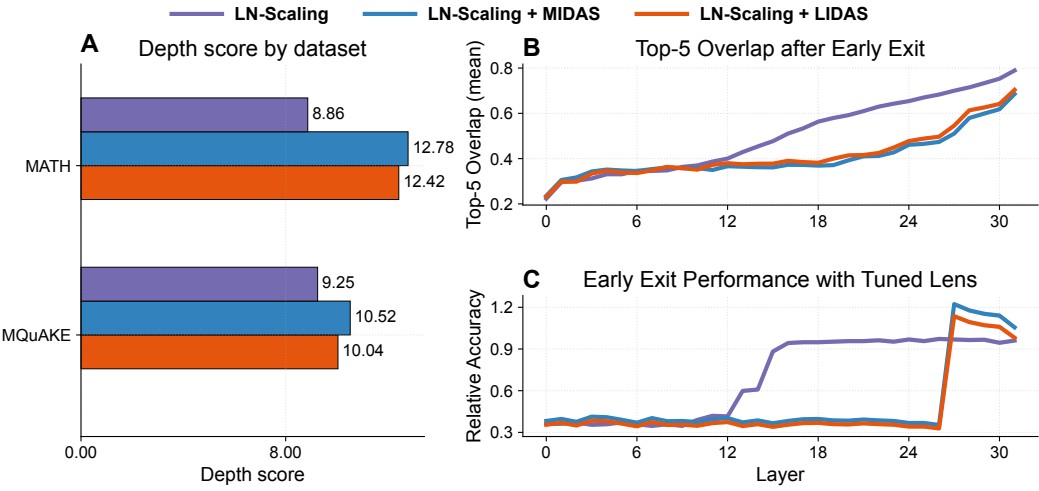

Figure 27: **Depth-grown models use their depth more also when applying growing techniques to `LN-Scaling` (360M)**

| | | Standard cooldown | | | | | | | Math cooldown | |
|---|---|---|---|---|---|---|---|---|---|---|
| | | **Holdout Set** (NLL ↓) | **Open-book Q&A** (F1 ↑) | **Closed-book Q&A** (F1 ↑) | **Lambada** (Acc ↑) | **Hellaswag** (Acc ↑) | **Math Word** (Acc ↑) | **Primitives** (Acc ↑) | **Math Word** (Acc ↑) | **Primitives** (Acc ↑) |
| 360M | LN-Scaling | **2.16** | 23.13 | **14.89** | 42.17 | 40.00 | 2.89 | 31.38 | 8.45 | 41.26 |
| | LN-Scaling + MIDAS | 2.19 | 22.04 | 14.05 | 42.77 | 39.84 | **4.03** | 33.04 | 7.72 | 36.90 |
| | LN-Scaling + LIDAS | **2.16** | 25.54 | 14.12 | **44.83** | **41.06** | 4.00 | **35.30** | **12.43** | **53.48** |
| 1.7B | LN-Scaling | 1.97 | **29.11** | **18.63** | 48.94 | 45.45 | 11.00 | **44.38** | 17.84 | 50.58 |
| | LN-Scaling + LIDAS | **1.96** | 28.04 | 18.42 | **51.37** | **46.69** | **17.32** | 43.32 | **23.98** | **56.28** |

Table 9: **Downstream performance of baseline and depth–grown models under `LN-Scaling`.** Compared to Table 1, `LIDAS` typically improves over the baseline, especially on reasoning-heavy tasks, but the gains are not uniform across datasets, indicating that growth combined with `LN-Scaling` yields architectures with qualitatively distinct behaviour.

## H    SCIENCE OF DL IMPROVEMENT CHALLENGE SUBMISSION

### H.1    WHAT MODEL ARE YOU TARGETING?

*Provide a summary of the problem the deep net model is designed to solve. Good summaries should outline the state of the literature, provide an overview that domain experts would consider reasonable, and cite relevant sources.*

We target pre-layernorm Transformer language models trained for autoregressive next-token prediction and used as general-purpose LLMs for language understanding and reasoning. A standard approach to increase model compute and capability is to scale depth (Kaplan et al., 2020; Hoffmann et al., 2022), yet a growing body of work shows that, in modern pre-layernorm Transformers, layers in the second half often have a disproportionately small influence on the final output distribution, also known as the *Curse of Depth* (Yin et al., 2024; Gromov et al., 2025; Li et al., 2025; Men et al., 2025; Sun et al., 2025; Csordás et al., 2025).

Our work focuses on training modifications that improve how Transformers use their depth. In particular, we study *gradual depth growth*, which starts from a shallow model and expands depth during training by duplicating and inserting layers, reusing learned weights to save compute (Gong et al., 2019; Reddi et al., 2023; Saunshi et al., 2024).

### H.2    HOW DO YOUR RESULTS CONTRIBUTE—OR COULD POTENTIALLY CONTRIBUTE—TO UNDERSTANDING THESE MODELS?

*What aspects of the models become better understood thanks to your work?*

Our results contribute to understanding *why* gradual depth growth can improve performance by directly linking it to depth-wise computation in Transformers. Using depth-aware diagnostics from recent work (Csordás et al., 2025) and Tuned Lens analysis (Belrose et al., 2023), we show that grown models rely on late layers substantially more than the non-grown baseline: early-exit predictions remain further from the final distribution and the depth score indicates more computation in deeper layers, particularly on reasoning tasks (Figure 1; Section 3.1). This provides evidence that depth growth counteracts the Curse of Depth by making later layers contribute meaningful new features rather than mostly small independent refinements.

Beyond *where* computation happens, we study *how* it is organized. Block-swap and skip interventions reveal that grown models form mid-network computational blocks that are comparatively robust to block-level reordering, indicating reduced layer-order dependence and the emergence of partially permutable computation (Figure 2a; Section 3.2). We further observe a repeating, block-wise pattern in attention contributions and in intervention effects, suggesting cyclical layer roles induced by the growth curriculum (Figure 2b; Section 3.3). Finally, comparing `MIDAS` and `LIDAS` isolates how subtle changes in the growth operator reshape internal structure: `LIDAS` yields more symmetric inter-block weight similarities and stronger engagement of middle-block attention sublayers, which correlates with its improved downstream results (Figure 6, Table 1).

### H.3    HOW DO YOU EXPECT YOUR SUBMISSION TO INFLUENCE FUTURE WORK?

*Propose ways in which your insights, findings, or methodologies could shape subsequent research directions, model design choices, or scientific applications.*

We expect our submission to influence future work in three actionable directions. First, it positions gradual depth growth as a principled training curriculum for mitigating depth underutilization, motivating further research on growth operators, growing schedules, and their interaction with normalization and optimization. Second, the diagnostics used here, depth score, tuned-lens early exit, and block-level interventions, offer a practical toolkit for evaluating whether proposed architectural changes truly induce deeper computation rather than superficial performance shifts, and could be adopted as standard analyses when scaling depth. Third, the emergence of permutable blocks and cyclical layer roles suggests opportunities to explicitly use the connection to *Looped* or *Universal* Transformers, which may improve robustness and enable reuse of learned computation to further increase depth for computation.

