# OpenReview forum: "Do Depth-Grown Models Overcome the Curse of Depth? An In-Depth Analysis"
_ICLR.cc/2026/Workshop/Sci4DL — Sci4DL 2026_

### Official Review · Reviewer_7CnR · 2026-02-23

**Fit:** 2
**Significance:** 2
**Confidence:** 2

**Summary:**

This paper investigates layer growth methods, one of which is a new method (LIDAS) introduced in this paper, and their impact on improving performance on simple reasoning tasks for transformers. The authors find that LIDAS performs similarly or better than alternatives. They also propose and validate various hypotheses for grown models, on relating deeper layer utilization to improved reasoning performance, and providing evidence for the emergence of distinct computational structures and patterns.

**Strengths:**

* Clearly written and well structured paper.
* proposes new method to grow models, that is similarly or more performant than alternatives
* Well chosen and varied experiments and methodology that builds a coherent picture
* Analysis conducted for two model sizes with consistent findings
* exploration and validation of multiple hypotheses
* Comparison to reasonable alternative models

**Suggestions:**

Regarding the current results, it would have been interesting to:
* see a comparison to LN-Scaling models, in addition to the baseline model, for Fig. 2 a and b.
* see an investigation into weight initialization effects for the reasoning performance of grown models, and the block structure / cyclical attention contribution.

Regarding the presentation of results:
* It's not immediately clear what standard and math cooldown mean. Which one is used for the rest of the results? I assume standard?
* A minor point, but I was initially confused by what you meant by blocks in section 3.2. This could be easily remedied by specifying that you are referring to blocks of layers. It becomes clear upon closer reading, or when checking the Appendix, but it would nevertheless decrease friction slightly.

The paper finds a lot of interesting and surprising results, which raise a lot of follow up questions and research venues:

* How are layer block structure and the associated cyclical pattern in attention related to reasoning ability?
* What role are attention heads playing within each layer block in grown models? Are attention heads in similar positions performing similar tasks across each block? Is the cyclical pattern associated with the increased reasoning performance, and if so how? This could follow the path of a more mechanistic analysis, where attention patterns across layers, in individual layer blocks, are investigated.
* Do these results hold and generalize for bigger models with more parameters than 1.7B?

---

### Meta-Review · Area_Chair_qciE · 2026-03-02

**Recommendation:** Accept

**Metareview:**

This work studies how growing the depth of transformers improves performance on simple reasoning tasks and introduces a new method. The methodology and the findings are a good fit for the workshop.

---

### Decision · Program_Chairs · 2026-03-02

Accept